A review of the impact of maize-legume intercrops on the diversity and abundance of entomophagous and phytophagous insects

http://orcid.org/0000-0001-7507-175X Pierre Jacques Fils 1 jacquesfilspierre@gmail.com
Jacobsen Krista L. 2
Latournerie-Moreno Luis 3
Torres-Cab Walther J. 3
Chan-Canché Ricardo 3
Ruiz-Sánchez Esau 3
1 Research Division, International Fertilizer Development Center , Muscle Shoals, Alabama , United States of America
2 Department of Horticulture, University of Kentucky , Lexington, Kentucky , United States of America
3 Division of Postgraduate Studies and Research, Tecnológico Nacional de México—Campus Conkal , Conkal, Yucatan , Mexico
Kumar Ravinder
Electronic publication date: 2023 Jun 26
Publication date: 2023
Volume: 11
Electronic Location ID: e15640
Received 2023 Mar 22; Accepted 2023 Jun 5
Copyright: © 2023 Pierre et al.
Copyright year: 2023
Copyright holder: Pierre et al.
License: This is an open access article distributed under the terms of the Creative Commons Attribution License, which permits unrestricted use, distribution, reproduction and adaptation in any medium and for any purpose provided that it is properly attributed. For attribution, the original author(s), title, publication source (PeerJ) and either DOI or URL of the article must be cited.
License URL: https://creativecommons.org/licenses/by/4.0/

Keywords: Maize, Legume, Intercropping, Entomofauna, Parasitoids, Predators, Biological control

Funding: National Council for Science and Technology (CONACYT) This study was funded by the National Council for Science and Technology (CONACYT) for their financial support through the Ph.D. CONACYT scholarship. The funders had no role in study design, data collection and analysis, decision to publish, or preparation of the manuscript.

==============================
In many parts of the world, chemical pesticides are the primary method of pest control in maize (Zea mays L.) crops. Concerns about the negative consequences of chemical pesticide use on people’s health and the environment, as well as the emergence of insecticide resistance, have accelerated attempts to discover alternatives that are effective, low-risk, and cost-effective. Maize-legume intercropping systems are known to have multiple benefits to agroecosystem functioning, including pest regulation. This review focuses on the influence of maize-legume intercropping systems on insect diversity and abundance as a mechanism for insect pest regulation in maize crops. First, this review combines knowledge of maize-legume intercrops, with a particular emphasis on the mechanism by which this practice attracts beneficial insects (e.g., predators, parasitoids) to reduce pest damage in intercropping systems. In addition, the pairings of specific legume species with the greatest potential to attract more beneficial insects and therefore reduce maize pests are also discussed. Finally, future research needs are also recommended. Findings are reviewed in the context of looking for long-term management strategies that can increase the adoption of integrated pest management programs in maize-based production systems.

Introduction

The need to increase global food production due to the rising human population is widely cited (e.g., Sachs et al., 2019; Tripathi et al., 2019) and is a key driver of intensification in agricultural production systems. Agricultural intensification through modern agricultural methods (i.e., utilizing contemporary technologies, techniques, and science to increase agricultural productivity) has led to landscape simplification and homogenization of biological and environmental structures (i.e., how all aspects of an ecosystem are functionally interconnected) in agroecosystems (Sachs et al., 2019; Dudley & Alexander, 2017; Sánchez-Bayo & Wyckhuys, 2019), increasing reliance on synthetic inputs and decreasing use of beneficial biotic interactions. The loss of natural habitats is leading to fragmentation, which is affecting biological regulations, and ultimately impacting the use of pesticides in agriculture. The widespread use and adoption of pesticides can be attributed to their ease of application and the immediate effects they have in providing effective crop protection against pests (Meynard & Girardin, 1991; Therond et al., 2017; Delecourt, Joannon & Meynard, 2019). Furthermore, pesticides are also favored for their ability to reduce the risk of limited or reduced yields (Ricciardi et al., 2021; Caron, Biénabe & Hainzelin, 2014; Cordell et al., 2011; Struik et al., 2014). This has led to a large decrease in agricultural biodiversity (Cabrera-Mireles et al., 2011), including insect diversity, which affects ecosystem services such as sustained soil fertility, decomposition, pollination, and biological pest control (Maldonado et al., 2008; Lozano & Jasso, 2012).

Preserving natural enemies in agroecosystems is vital for strengthening biological pest control services (Landis, Wratten & Gurr, 2000; Barbosa, 2003). For example, conservation biological control utilizes existing agent species in a region and relies on habitat management to provide ecological resources to natural enemies or have direct effects on pests, rather than importing or releasing new species (Poveda, Gomez & Martinez, 2008; Letourneau et al., 2011; Lu et al., 2014). In conservation biological control methods, increased plant diversity frequently leads to an increase in species diversity and quantity of beneficial insects, including natural enemies (Langellotto & Denno, 2004; Berndt, Wratten & Scarratt, 2006; Letourneau et al., 2012). This can be achieved by influencing the ecology that foster the growth and survival of natural enemies (Altieri & Letourneau, 1982; Andow, 1991; Landis, Wratten & Gurr, 2000). Some practices within fields, as well as surrounding landscape structure and composition, have been widely shown to affect the diversity and abundance of natural enemies in agroecosystems (Colunga-Garcia, Gage & Landis, 1997; Bommarco, 1998; Elliott et al., 1999; Tscharntke et al., 2005). Increased plant diversity in agroecosystems can enhance pest management through various mechanisms. The “Enemies Hypothesis” suggests that greater plant diversity can increase the number of natural enemies of herbivores by providing more nutrients. The “Resource Concentration Hypothesis” suggests that specialist herbivores may have difficulty locating and competing for host plants in a diverse system, resulting in fewer pests. Additionally, plant diversity can improve conditions for natural enemies by providing alternative food sources, shelter, and increasing prey availability (Root, 1973; Landis, Wratten & Gurr, 2000).

Intercropping is an agroecological practice that promotes plant diversity within crop fields, which may enhance conditions for natural enemy populations (Gontijo, 2019; Zhou et al., 2014). It involves planting two or more crops simultaneously in the same field, and these diversified agricultural systems can improve productivity and ecosystem services, including biological control (Beaumelle et al., 2021; He et al., 2019). In regions such as Africa and the Americas, intercropping maize with legumes, such as common beans, groundnuts, cowpea, and lima beans, is a traditional practice that has gained popularity in modern agroecological farming systems (Tsubo & Walker, 2004; Pierre et al., 2021; Pierre et al., 2022b). Cereal-legume intercropping has been documented to reduce the abundance of specialist pests and provide additional resources, such as shelter and food, for arthropod predators (Norris et al., 2018). The “Resource Concentration Hypothesis” proposes that specialist herbivores may have difficulty identifying and competing on their host plants within a complex of diverse plants, thereby lowering pest populations in polycultures (Root, 1973). Increasing crop diversity also lessens the impact of pest outbreaks (Sekamatte, Ogenga-Latigo & Russell-Smith, 2003) and boosts the population of generalist predators, ultimately reducing pest insects (Heimpel & Jervis, 2005). Maize-legume intercropping is widespread in Africa, Latin America, and parts of Asia, while it is becoming more common in Europe and North America (Canada and the United States of America) (Li, Zhang & Zhang, 2013; Segura et al., 2022). Overall, intercropping can contribute to the sustainability of agriculture by promoting a more diverse and balanced agroecosystem.

In the development of more sustainable agricultural systems, crop management strategies that attempt to reduce chemical inputs are crucial. In the context of biological control, detailed maize-legume intercropping reviews are required to improve understating of the mechanisms by which this system may increase the diversity of beneficial insects, thereby reducing the need for chemical insecticides. Specifically, a comprehensive evaluation of the effect of maize-legume intercropping on insect diversity and abundance can strengthen our understanding of the potential of this approach to improve biological pest control (Gontijo, 2019; Marrec et al., 2015).

This work is significant as in tropical areas, intercropping is widely used by small-scale farmers. It is also relevant for researchers due to the lack of information about the mechanism used by this practice to control major pests in the main crop (i.e., maize) and therefore reduce the use of insecticides in agriculture. Although intercropping is a widespread traditional agricultural practice world-wide, to the best of our knowledge, few studies have conclusively demonstrated how maize intercropped with legumes increases the population of beneficial insects, resulting in less phytophagous damage in maize plants.

Based on that, we hypothesized that where greater numbers of flowering legume species were intercropped with maize, larger populations of beneficial insects would be encountered, thereby reducing the population of phytophagous insects in maize-based intercropping systems. However, this relationship has not been robustly reviewed in the context of this traditional farming practice. Therefore, this review explores the role of entomofaunal diversity in controlling agricultural pests in maize-legume intercropping systems and investigates the mechanism by which these natural enemies control pest populations.

Survey methodology

This article presents a comprehensive bibliographic literature review conducted between September and December 2022 to examine the impact of maize-legume intercrops on entomophagous and phytophagous insect diversity and abundance, as well as the benefits of intercropping maize with legume crops to enhance biological control. The search included English language resources, such as journals, reports, articles, and proceedings, from the past 5 years, and earlier studies were also included in areas where current publications were lacking (Fig. 1). To identify relevant resources, we employed the search terms “intercropping” AND “maize” OR “corn” AND “predators” OR “parasitoids” AND “phytophagous insects” on Google Scholar and Web of Science, resulting in the identification of 80 resources meeting our inclusion criteria (Fig. 1). The publications were saved as plain text files containing full citation data for further analysis. A preliminary analysis was conducted using Microsoft Excel and Analyze Results to examine annual publications, authors, journals, countries, institutions, and disciplinary distribution. Publications that did not address the research questions or did not provide an overview of the benefits of maize-legume intercropping were excluded. The selected studies were analyzed for relevant data on insect diversity and abundance, as well as biological control in maize-legume intercropping systems. The information was categorized and subcategorized, including maize-legume intercrop (predators, parasitoids, and phytophagous insects). Graphs were generated using Microsoft Excel to visualize the results.

Figure 1 The distribution of publications on the impact of maize-legume intercrops on insect diversity and abundance over time, from 1961 to 2022.

The importance of entomofauna in maize-legume agroecosystems

Entomofauna in these systems includes a large number of “phytophagous” species (i.e., phytophagous) that cause crop damage, as well as beneficial insect species (e.g., predators, parasitoids) capable of suppressing phytophagous species populations. In cereal-legume agroecosystems, maize-legume intercrops are a good model system that can help us better understand the behavior of insects and their associated natural enemies in such cropping systems (Zou & Li, 2002). For example, several studies have reported that maize intercropped with legumes had the greatest diversity and abundance of natural enemies, including a number of parasitoids known to attack S. frugiperda (J. E. Smith), a Lepidopteran pest attributed to up to 60% yield damage in some maize-producing regions globally (Jiménez-Martínez & Gómez-Martinez, 2013; Goergen et al., 2016). Similar results were also reported by Scheidegger et al. (2021). To address the likelihood that phytophagous insects would spread into new areas and destroy maize crops in intercropping systems, a comprehensive study of the effect of maize-legume intercrops on the diversity and abundance of entomofauna is needed to create techniques that could be used to reduce the potential for pest infestations.

Phytophagous insects damage in maize-legume systems

The two most important insect pests of corn in tropical regions are the fall armyworm S. frugiperda, which primarily causes damage by feeding on both vegetative and reproductive plant parts, and corn earworm (Helicoverpa zea (Boddie, 1850) (Lepidoptera)), which damages ears by feeding initially on silk tissue and then on the ear tip and kernels (Jones et al., 2019; Niassy et al., 2021). S. frugiperda is a polyphagous insect that causes substantial losses not only in maize but also in other economically significant cultivated species such as sorghum, rice (Oryza sativa L.), soybean (Glycine max L.), and cotton Gossypium hirsutum L. (He et al., 2021), as well as in the fruit, resulting in premature drop and fruit rot on tomatoes and peppers (Abrahams et al., 2017). Uncultivated plant species from the Poaceae, Asteraceae, and Fabaceae families also serve as hosts for S. frugiperda (Montezano et al., 2018). S. frugiperda can cause yield losses of up to 60% in some maize-producing regions (Goergen et al., 2016). In recent years, S. frugiperda has caused extensive damage to maize in Africa and India (Goergen et al., 2016), affecting both large-scale and small-scale farmers.

The larvae of Helicoverpa zea can also cause harm by burrowing into maize tassels and ears (Abrahams et al., 2017). Young larvae hide in the corn funnel during the day and eat leaves at night (Day et al., 2017). Corn earworm can infest over 20 cultivated species apart of corn, but it predominantly feeds on cotton, soybean, tomato, sorghum, and wild plants (Sosa-Gómez et al., 2016). This insect can damage 61% to 100% of the ears and cause up to 10% maize grain yield loss, with yellow grain varieties suffering the most (Rodriguez-del-Bosque, Cantú-Almaguer & Reyes-Méndez, 2012). Variations in regional temperatures and precipitation can lead to a large range extension of corn earworm (Jones et al., 2019). Other pest insects include stem maggots (Ophiomyia phaseoli (Tryon) (Diptera: Agromyzidae)), ootheca (Ootheca benigeni (Coleoptera: Chrysomelidae)), bruchids (Acanthoscelides obtectus and Zabrotes subfasciatus (Coleoptera: Chrysomelidae)), and aphids (Aphis fabae (Hemiptera: Aphididae)) cause 37% to 100% yield loss in common bean. Additionally, pests such as corn rootworms or cucumber beetles (Diabrotica spp.) pose a significant threat to maize crops as they feed on the roots of young plants. Other pests that affect maize production include the European corn borer (Ostrinia nubillais), the corn leaf aphid (Rhopalosiphum maidis), and cutworms (Agrotis ipsilon).These species also consume a variety of economically important crops like cowpea, lima beans, common bean, soybean, maize, and peanuts (Arachis hypogaea L.) (Laumann et al., 2003). In the initial phase of the crop (1 week after emergence), adults and larvae can cause defoliation rates as high as 16% in 24 h, which significantly reduces photosynthetic capacity and competitiveness of leguminous plants like cowpea and lima beans (Silva et al., 2003). This damage exceeds the economic damage caused by other defoliating insects (Silva et al., 2003). These authors suggest controlling the insect in its earliest stages, when only two insects per plant are present.

In maize-legume intercrops, legume pests can also be affected by the system (Bukovinszky et al., 2004). Bean infestation with black aphids (Aphis fabae) was considerably lower when beans were intercropped with older and taller maize plants (Ogenga-Latigo, Ampofo & Baliddawa, 1992). There were fewer insects on the cowpea crop when it was cultivated in combination with maize at certain ratios, compared to when it was produced in monoculture (Olufemi, Pitan & Odebiyi, 2001). In addition, Sekamatte, Ogenga-Latigo & Russell-Smith (2003) discovered that intercropping maize with groundnut, soybean, and common bean considerably reduced termite attack and resultant loss in grain yield of maize compared to maize as the single crop, while increasing the number of predatory ants in maize fields. Although termites are not a problem for maize in the Americas (as they typically feed on dead wood), in Africa various species of termites (A. evuncifer and C. sjostedti) have been reported as highly damaging species to maize (Loko et al., 2021). Also, peanuts and soybeans were more efficient than ordinary beans in preventing termite infestation, indicating the need to choose optimal legumes for each intercropping condition (Sekamatte, Ogenga-Latigo & Russell-Smith, 2003).

Cereal-legume intercropping can affect insect diversity and abundance by increasing foliage nitrogen content, altering palatability to herbivores. An intercropping partner boosts predation rates, but high nitrogen levels reduce them, disappearing at maximum levels (Wang et al., 2021). Intercropping collard with beans or onions suppresses diamondback moth larvae populations, but nitrogen application may increase damage without affecting larval numbers (Said & Itulya, 2003). Higher nitrogen rates may cause increased insect damage in sorghum (Ampon’g-Nyarko & Nyang’or, 1991). Fertilization affects host plant nitrogen content insignificantly, altering pea aphid reproduction (Buchman & Cuddington, 2009). Nitrogen-fertilized Medicago truncatula produces heavier and more fecund pea aphids (Gao et al., 2018). Nitrogen input may not affect alfalfa’s phloem sap, pea aphids’ food source, but requires experimental verification (Phillips, 1980; Buchman & Cuddington, 2009). Excessive nitrogen may not be absorbed or used by alfalfa, but within a specific range, it enhances alfalfa’s resistance to pea aphids (Maltais & Auclair, 1957). Further research is needed to determine the mechanisms underlying the negative correlation between nitrogen inputs and pest population growth.

In this view, the presence of legumes in intercropping systems might boost the abundance and effectiveness of natural enemies, whose resource needs are dependent on habitat complexity. Adding more plant species can change the environment and the quality of the host plant. This can have a direct effect on how herbivorous insects look for host plants and indirect effects on how fast they grow. Despite this, information on the strategies utilized by phytophagous insects in maize-legume intercropping systems to resist harm from beneficial insects was found to be lacking.

Predator diversity and abundance

In maize-legume intercropping systems, it is well established that crop diversification impacts the populations of insect predators (Maitra et al., 2019). One mechanism that increases the presence of intraguild predators in ecosystems is attributed to greater plant diversity and a wider variety of herbivores to eat them (Tixier et al., 2013).

Multiple studies have demonstrated that maize-legume intercrops may attract and harbor generalist predator populations (Table 1). For example, Kebede et al. (2018) discovered that the abundance of generalist predators tended to be lower in maize monocultures than in maize intercropped with legumes. Similarly, the abundance of coccinellid predators and geocorid bugs was significantly higher in maize intercropped with groundnut compared to the sole crop of maize (Udayakumar, Shivalingaswamy & Bakthavatsalam, 2021). However, these benefits may not be consistent for some specialist predators. For example, it has been reported that some specialist predators that help suppress key agricultural pests find their prey more quickly when the environment in which they are searching is uniform.

Table 1 Literature documenting the effect of maize-legume intercropping on the abundance of predators.

Intercropping system	Effects	Author	
Maize intercropped with groundnut	Increased the abundance of coccinellid predators and geocorid bugs compared to the sole crop of maize.	Udayakumar, Shivalingaswamy & Bakthavatsalam (2021)	
Maize-legume (e.g., beans, cowpea, groundnut)	Boosted the population of predators and parasites, which, along with the diverse range of crops, assisted in maintaining pest damage below the economic threshold.	Maitra et al. (2019)	
Maize-cowpea	The number of predators increased, but there was no statistical difference between the intercropping system and the maize monocrop.	Pierre et al. (2022a)	
Maize-crotalaria	No statistically significant difference was observed between the intercropping and monocropping systems in terms of predator diversity.	Pierre et al. (2022a)	
Maize, cowpea, and squash	No significant effect on the number of predators was observed comparable to those of maize grown as a monocrop.	Letourneau (1987)	
Maize-groundnut-soybean-common bean	Intercropping reduced termite attack and increased predatory ants in maize fields, with peanuts and soybeans being more efficient than ordinary beans.	Sekamatte, Ogenga-Latigo & Russell-Smith (2003)	

In addition, even though the provisioning of floral resources and habitat in maize-legume intercropping can increase the density and diversity of natural enemies (Root, 1973; Pimentel, 1961), it seems unlikely that crop diversification alone increases the effectiveness of natural enemies. Furthermore, not all legume species are capable of attracting beneficial insects to intercropping systems (Sekamatte, Ogenga-Latigo & Russell-Smith, 2003). Peanut and soybean floral resources are more successful at preventing termite infestations than common bean floral resources but can be detrimental to natural enemies (Goleva & Zebitz, 2013). For instance, particular flower pollen is harmful to predatory mites (Goleva & Zebitz, 2013). In addition, Baggen, Gurr & Meats (1999) found that extrafloral nectaries of dill, buckwheat, and faba beans were beneficial to Copidosoma koehleri and Phthorimaea operculella Benth. To provide nectar and pollen, “proper” floral host plants should have blooming phonologies that coincide with the seasonality of a given predator (Andow, 1991; Altieri & Letourneau, 1982). Some of the earliest and most influential research on intercropping with field mustard (Brassica campestris) revealed that natural enemies were more numerous and diversified in monocultures than in stands of mixed crop species (Root, 1973; Pimentel, 1961). More recent work has suggested that careful consideration must be given to where a legume intercrop is sown (e.g., between rows as opposed to between planting stations) as the arrangement of the crops may affect directly the predator insects attraction (O’Rourke et al., 2006, Weber, 1996).

Other mechanisms can also contribute to the increase in predator insects in intercropping systems. Some of these, for example, use plant odors or visual cues to locate their hosts (Mohler & Johnson, 2009). In some circumstances, intercropping can interfere with the predator finding the prey (Smith & McSorley, 2000). To enhance the number of beneficial insects, the intercrop should provide them with important resources, such as pollen, nectar, alternate prey, shelter, or overwintering locations (Mohler & Johnson, 2009).

Changes in the environment and the quality of the host plant also have direct implications on the host plant-seeking behavior of herbivorous insects, as well as indirect effects on their developmental rates and interactions with natural enemies such as predators (Ansari et al., 2011). The environment of the host plants, such as adjacent plants and microclimatic circumstances, is altered first, followed by the host plant’s shape and chemical composition (Langer, Kinane & Lyngkjsr, 2007). In addition, the production of volatiles and the release of additional floral nectar enable plants interactions with natural enemies of insect pests (i.e., predators), which actively limit the number of herbivorous insects (Dudareva et al., 2006; Maffei, 2010). Despite the fact that maize-legume intercrops can reduce maize pest infestation, specific studies on how some generalist predators can have a negative impact on the diversity and abundance of other predators have been understudied.

Parasitoid diversity and abundance

For the control of S. frugiperda, planting leguminous intercrops is a strategy that is employed in traditional and conservation agricultural systems around the world (Harrison et al., 2019). Harrison et al. (2019) found that intercrops or border crops give natural enemies olfactory clues, which causes S. frugiperda to be naturally parasitized. When under herbivore attack, numerous plants release volatiles that serve as prey/host location cues for predators and parasitoids (Turlings & Benrey, 1998; Dicke & Vet, 1999). For example, the (E)-B-caryophyllene generated by maize roots in response to Diabrotica virgifera (corn rootworm) attracts entomopathogenic nematodes in maize plants (Rasmann et al., 2005). Increasing the release of these volatiles could enhance the effectiveness of these natural enemies (Kessler & Baldwin, 2001). In maize, caterpillar feeding induces a release of a volatile mixture that is enticing to numerous parasitoids (Harrison et al., 2019).

In maize-legume intercropping systems, Maitra et al. (2019) demonstrated that this system can increase the populations of beneficial insect parasitoids (Table 2). Significantly higher rates of parasitization of egg masses of S. frugiperda by Trichogramma sp. (Hymenoptera: Trichogrammatidae) were recorded in maize/broad bean intercrops compared to the monocrop of maize (Udayakumar, Shivalingaswamy & Bakthavatsalam, 2021). Parasitoids readily attack the larvae and have the potential to reduce and damage the S. frugiperda populations in maize plants (Bianchi, Booij & Tscharntke, 2006). Hailu et al. (2018) found that intercropping maize with leguminous crops lowers S. frugiperda damage, notably in the early growth phases of the maize up to tasseling. This may be because some legume species in the intercropping system can provide nectar, pollen, shelter, and chemical protection for resident parasitoids, for example, floral volatiles produced by cowpea flowers (Dannon et al., 2010), thereby encouraging parasitoid population growth in maize and legume intercropping systems (Day et al., 2017). High numbers of parasitoids were recorded in push–pull and maize-cowpea intercrop plots as opposed to maize sole cropping systems (Mudare et al., 2022). Other research found that a large number of parasitoids in maize-legume intercropping systems correlates with a high level of biodiversity (Afrin et al., 2017).

Table 2 Literature documenting the effect of maize-legume intercropping effects on the abundance of parasitoids.

Intercropping system	Effects	Author	
Maize intercropped with broad bean	Significantly enhanced rates of egg mass parasitization by Trichogramma sp. compared to maize grown as monocrop.	Udayakumar, Shivalingaswamy & Bakthavatsalam (2021)	
Maize intercropped with cowpea	A greater number of parasitoids were observed compared to a maize monocrop.	Pierre et al. (2022a)	
Maize intercropped with crotalaria	Increased the diversity of parasitoid insect compared to sole maize.	Pierre et al. (2022a)	
Maize-cowpea-squash intercrops	Exhibited a higher percentage of egg and larval parasitism compared to maize monocropping systems.	Letourneau (1987)	
Maize intercropped with other legumes (e.g., soybean, cowpea)	In this study, it was observed that intercrops had up to two-fold greater levels of egg parasitism by scelionid Telenomus spp. than monocrops.	Chabi-Olaye et al. (2005)	

Similar to entomophagous predator populations, maize-legume intercrops have been linked to different benefits for attracting generalist parasitoids. However, this may not be the case for some specialist parasitoids when it comes to the ability of maize-legume intercrops to attract generalist parasitoids. Some of the earliest and most influential research on intercropping with other types of crops revealed that natural enemies were more abundant and diversified in monocultures than in intercropping systems (Root, 1973; Pimentel, 1961). Moreover, despite the fact that the provision of floral resources and habitat in maize-legume intercropping can increase the density and diversity of natural enemies (e.g., parasitoids) (Heimpel & Jervis, 2005), crop diversity likely does not increase the effectiveness of natural enemies. As with predators and other beneficial insects, parasitoids can also be anthophilous. Therefore, it is important to ascertain which legume species are required for maize-legume intercropping systems to offer the essential resources to attract parasitoid insects. For example, in maize-legume intercropping systems, peanut and soybean floral resources were found to be more attractive to beneficial insects than common bean floral resources. However, some natural resources can be detrimental to natural enemies (Mudare et al., 2022). In order to deliver nectar and pollen, “suitable” floral hosts should have blooming phenologies that correlate with the seasonality of a certain parasitoid. This has been previously well-documented in the literature (e.g., Altieri & Letourneau, 1982; Andow, 1991).

The increase in parasitoid species number can be linked to multiple causes in intercropping systems. Some of these insects use plant scents or visual cues to locate their hosts by locating the plants on which they are located (Mohler & Johnson, 2009). In such instances, intercropping can impede the ability of parasitoids to locate hosts (Smith & McSorley, 2000). In order for the intercrop to increase the population of beneficial insects, it must provide pollen, nectar, alternative prey, shelter, and overwintering sites (Mohler & Johnson, 2009). Furthermore, by reducing the spread of disease-carrying spores and modifying the environment to be less favorable to the development of certain infections, related crop species can help delay the onset of diseases (Langer, Kinane & Lyngkjsr, 2007) The increase in parasitoids in maize-legume intercrops can also be explained by the volatile chemicals emitted by companion crops in intercrops. In the olfactometer bioassay, Cotesia icipe (Hymenoptera: Braconidae) and Coccygidum luteum, two major larval parasitoids of S. frugiperda, were attracted to the volatiles of all three companion plant species (Sobhy et al., 2022). Despite being described as one of the key possibilities to reduce maize pest infestation in intercrops, maize-legume intercrop systems lack specialized research on how parasitoids deal with various resistant-phytophagous insects.

Future prospects

Including legume plant species in maize-based intercropping systems is regarded as a highly effective and dynamic insect pest control tactic. Nevertheless, the selection of legume species with sufficient mass-flowering must be considered to assure the effectiveness of maize-legume intercrops to attract more beneficial insects and therefore reduce the pest population in maize. It is vital to comprehend the process by which maize-legume interactions increase insect diversity and abundance in order to develop effective pest management techniques for maize-legume intercropping systems around the world. Extremely sophisticated phytophagous insect defenses against beneficial insects have impeded the selection of insect-resistant legumes for intercropping with maize-based planting strategies; however, this is not always the case as suggested by the “resource concentration” hypothesis. This review demonstrated that maize-legume relationships boosted the diversity and abundance of beneficial insects, consequently decreasing the number of phytophagous insects.

Although maize-legume intercropping systems have been identified as a promising approach to reduce pest infestation in maize, there is a lack of specialized research on how parasitoids and predators interact with various resistant phytophagous insects in these systems. Future research should focus on discovering how insect pests have co-evolved to resist the abilities of beneficial insects to reduce them. Identifying the mechanism of phytophagous insect counter-adaptations to natural enemies would allow us to comprehend the rate at which maize insects adapt to natural enemies and set new objectives for the development of sustainable pest control programs in maize-based legume intercropping systems. Further, the discovery of legume species that attract particular predators and parasitoids may play a crucial role in boosting the sustainability of pest management programs in maize-based intercropping systems. For example, Sekamatte et al. (Brennan, 2016) discovered that maize intercropped with soybean and groundnut was more efficient than ordinary beans in controlling termite attack, implying the importance of identifying optimal legumes for each cropping circumstance. Lastly, knowing how maize-legumes might protect themselves from damage by phytophagous insects could help us understand how beneficial insects, phytophagous pests, and plants interact with each other in a way that is always changing.

Conclusions

The findings of this review demonstrate that intercropping maize and legumes may increase insect diversity and abundance and that beneficial insects preferred maize-legume intercropped habitat by altering the population dynamics of insects in the cropping system. In addition to volatile compounds emitted by partner crops in intercropping, the maize-legume intercropping system enhances the number of beneficial insects by providing pollen, nectar, alternate prey, shelter, and overwintering places. Thus, the maize-legume intercropping system promotes an environment suitable for the optimal delivery of ecosystem services, which may contribute to agricultural sustainability. Nevertheless, there is ample room for additional research on interactions between phytophagous insects and beneficial insects in maize-based intercropping systems, particularly identifying which legume species is more likely to attract more effective beneficial insects that can harm maize phytophagous pests. Even though maize-legume intercropping is one of the most effective ways to eliminate maize pests, little is known about how generalist predators can have a negative effect on the diversity of other predators and how phytophagous insects in maize-legume intercropping systems protect themselves from predators and parasitoids. This opens the door for additional research on the influence of maize-legume intercropping on the interaction between insect-plant relationships. Overall, intercropping maize with legumes can be used to effectively control pests in maize production systems, hence reducing the requirement for chemical insecticides.

Additional Information and Declarations

Competing Interests

Author Contributions

Data Availability

Jacques Fils Pierre is employed by the International Fertilizer Development Center. The authors declare that they have no competing interests.

Jacques Fils Pierre conceived and designed the experiments, performed the experiments, analyzed the data, prepared figures and/or tables, and approved the final draft.

Krista L. Jacobsen performed the experiments, analyzed the data, authored or reviewed drafts of the article, and approved the final draft.

Luis Latournerie-Moreno analyzed the data, authored or reviewed drafts of the article, validation, and approved the final draft.

Walther J. Torres-Cab analyzed the data, authored or reviewed drafts of the article, project administration, and approved the final draft.

Ricardo Chan-Canché analyzed the data, authored or reviewed drafts of the article, project administration, and approved the final draft.

Esau Ruiz-Sánchez conceived and designed the experiments, performed the experiments, analyzed the data, authored or reviewed drafts of the article, and approved the final draft.

The following information was supplied regarding data availability:

This is a literature review.

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
