# Peer review of "A review of the impact of maize-legume intercrops on the diversity and abundance of entomophagous and phytophagous insects"

_PeerJ, doi:10.7717/peerj.15640_

## Round 0.1 · original submission · Major Revisions

The manuscript is required major revisions. Authors are requested to address the reviewers' comments carefully to make the manuscript acceptable.

·

Basic reporting

I think the article meets the standards 1-3 of the Journal

Experimental design

The methods are sound

Validity of the findings

The paper reviews a topic that is important for pest management. I found the review well prepared overall.

Additional comments

Fils Pierre at al., revised the impact of maize-legume intercrops on the diversity and abundance of insects. The authors used 80 papers from various sources of literature. Overall, I found the review well-prepared and well written. A have a few suggestions addressed below:

Main suggestions:
-I think the authors may be missing an important aspect of maize-legume intercropping. We know that legumes have nitrogen-fixing bacteria that improves soil nutrition to other crops. This increase of nitrogen may also have an effect on insect diversity and insect abundance when they consume foliage. I suggest searching the literature and include either a section or a paragraph about this.

-Lines 156-187: Section “Phytophagous insects damage in maize-legume systems” The authors mention a few important herbivore pests of maize but are missing a few important ones that should be included: these are: European corn borer (Ostrinia nubillais), the corn leaf aphid (Rhopalosiphum maidis), and cutworms (Agrotis ipsilon)

Minor suggestions:
Line 142: I think removing the word “also” will make the sentence flow better.
Line 171: a part should be “apart”
Line 186: Controlling the crop? Or controlling the insect?
Line 194: The authors talk about termite attack to corn in several parts of the manuscript. Termites aren’t a problem of maize in the Americas (they usually feed on dead wood), it would be helpful to include a sentence stating that they are a problem (not sure how common) in Africa, so that the reader understand the context.
Table 1: second row column 1 (Maize-legume): which type of legume (Maitra et al 2019)? Same in “maize-legume-squash” three rows below. Which legume?
Table 2: similar comment as above “Maize intercropped with other legume” which legume?
Lines 345-348: These two sentences are a bit unclear. Are the authors stating that insects have evolved resistance to predators and parasitoids? Please elaborate more on what is know about these mechanisms.
Line 393: State which author obtained the PhD scholarship.

Reviewer 2 ·

Basic reporting

The ms submitted proposes to focus on maize-legume intercropping systems and the effect on insect diversity and abundance. The authors suggest that this is "a mechanism for insect pest regulation in maize crops". Although the topic could be of interest for the reader and the proposed review would be a contribution, it fails in many aspects, being the main concern the lack of information and proposed standards for meta analyses and systematic reviews proposed in the literature. For example see PRISMA Group (Moher et al. 2009). At the most it is a report of studies without a quantitative aspect, but more concerning is that the terms of inclusion and inclusion of the studies are not clearly stated, being rather difficult to weight the support for their conclusions.

Experimental design

Fails to clearly describe and include the requirements for carrying out a systematic review I would suggest te authors read
Moher D, Liberati A, Tetzlaff J, Altman DG, PRISMA Group (2009) Preferred Reporting Items for Systematic Reviews and Meta-Analyses: The PRISMA Statement. PLoS Medicine 6:e1000097. doi:10.7326/0003-4819-151-4-200908180-00135


Nakagawa S, Noble DWA, Senior AM, Lagisz M (2017) Meta-evaluation of meta-analysis: ten appraisal questions for biologists. BMC Biol 15:18. doi:10.1186/s12915-017-0357-7

Validity of the findings

As the above has not being carried out under the accepted standards, the finding and conclusions are rather questionable in the ms present state, but could be improved, if following the above comments and re submitted as a new paper.

Additional comments

other suggestions

line 49 a gap
line 54 "For example, conservation biological control consists in modifying habitats to support and preserve the population of native beneficial insects (i.e. parasitoids and predators) for effective biological control of crop pests (Bone et al., 2009)." this a recurrent misunderstanding of the definition. I would suggest you read for the original source. CBC does entail habitat manipulation but not any type o HM, and includes reduced pesticide insects mortality in this original definition (See Wartten for example)
Survey methodology
Survey methodology si poorly described, the sources of information are mentioned but no exclusion terms stated. Some inclusion terms are. No quantitative information is given as the result of the revision (systematic ?)

conclusion are not supported by data

tables have data from 6 and 5 papers...what about the other 74-papers that were reviewed?
both tables are not references in the general text

Reviewer 3 ·

Basic reporting

Dear Authors and Editor,

I am writing in reference to manuscript #83630 entitled "A review of the impact of maize-legume intercrops on the diversity and abundance of entomophagous and phytophagous insects", in which a bibliographical review is carried out on the influence of maize-legume intercrops on the abundance of insects. In this sense, intercropping practices are being strongly promoted to modify through technical aspects the sustainability problems of the dominant agricultural systems, although we must emphasize that it is a historical practice that was lost due to the specialization of agroecosystems. The text presents relationships that we could discuss: “Natural habitats are becoming increasingly fragmented due to the loss of habitats, resulting in a loss of biological regulations, making the use of pesticides inevitable” (line 48 to page 49), I do not think that this relationship is so linear. The use of pesticides, and their adoption, beyond extension systems that promoted (and mostly promote) systems based on chemical inputs (Therond et al. 2017), is the ease and instantaneity of their effects.

In my opinion, the definition of the hypotheses of the enemies, and the concentration of resources is confusing, I should rewrite these paragraphs (line 64 to page 96). Remember that scientific names must have order and family the first time they are named in the text “Other pest insects include stem maggots (Ophiomyia phaseoli), ootheca 177 (Ootheca benigeni), bruchids (Acanthoscelides obtectus and Zabrotes subfasciatus), and aphids 178 (Aphis fabae) cause 37% to 100% yield loss in common bean”,(page 176 to page 179), for example Ophiomyia phaseoli (Diptera: Agromyzidae).

Experimental design

For the materials and methods section (page 125 to page 137) you would primarily need to include an annex with the references of all documents included in the manuscript. More relevant than this is to save the problem that the sampling methodology is explained in a very imprecise way, you must rewrite the entire section to know how to do it and be able to replicate the bibliographic review. It should include the search string, the explicit protocol followed, as well as the years included in the search (in the text it only suggests "the last 5 years"). In addition, it would be interesting to see in a figure the temporal trend of works published on the topic.

As an opinion, but to be evaluated by the authors…perhaps a map that summarizes tables 1 and table 2? since both the system and its effects of each particular work appear in the text. In this sense, I would expect the authors to thoroughly review the items included here to be considered for publication.

Validity of the findings

no comment

Additional comments

Attached below some references to keep in mind. I hope that the comments made here will serve to enrich the submitted manuscript,

Best regards.


Therond, O., Duru, M., Roger-Estrade, J., & Richard, G. (2017). A new analytical framework of farming system and agriculture model diversities. A review. Agronomy for sustainable development, 37, 1-24.

Rhainds, M., & English‐Loeb, G. (2003). Testing the resource concentration hypothesis with tarnished plant bug on strawberry: density of hosts and patch size influence the interaction between abundance of nymphs and incidence of damage. Ecological Entomology, 28(3), 348-358.

Gonzalez, E., Landis, D. A., Knapp, M., & Valladares, G. (2020). Forest cover and proximity decrease herbivory and increase crop yield via enhanced natural enemies in soybean fields. Journal of Applied Ecology, 57(11), 2296-2306.

Dainese, M., Martin, E. A., Aizen, M. A., Albrecht, M., Bartomeus, I., Bommarco, R., ... & Steffan-Dewenter, I. (2019). A global synthesis reveals biodiversity-mediated benefits for crop production. Science advances, 5(10), eaax0121.

---

## Round 0.2 · accepted · Accept

The authors have satisfied the reviewers' queries, hence it is accepted. Authors may change one sentence in proof

Line 221: "Fertilization affects host plant nitrogen content insignificantly..." Do the authors mean that fertilization does not affect plant nitrogen content?

·

Basic reporting

The authors have addressed my comments. There is one sentence that I'm concerned about and would like the authors to revise.
Line 221: "Fertilization affects host plant nitrogen content insignificantly..."
Do the authors mean that fertilization does not affect plant nitrogen content?

Experimental design

Looks good to me

Validity of the findings

They are good to me